# Diode-Like Current Leakage and Ferroelectric Switching in Silicon SIS Structures with Hafnia-Alumina Nanolaminates

**DOI:** 10.3390/nano11020291

**Published:** 2021-01-22

**Authors:** Vladimir P. Popov, Fedor V. Tikhonenko, Valentin A. Antonov, Ida E. Tyschenko, Andrey V. Miakonkikh, Sergey G. Simakin, Konstantin V. Rudenko

**Affiliations:** 1Rzhanov Institute of Semiconductor Physics SB RAS, 630090 Novosibirsk, Russia; ftikhonenko@gmail.com (F.V.T.); ava@isp.nsc.ru (V.A.A.); tys@isp.nsc.ru (I.E.T.); 2Valiev Institute of Physics and Technology RAS, 117218 Moscow, Russia; miakonkikh@ftian.ru (A.V.M.); simser@mail.ru (S.G.S.); rudenko@ftian.ru (K.V.R.)

**Keywords:** SIS structures, silicon-on-ferroelectric, diode and FET characteristics, leakage mechanisms, ferroelectric hysteresis

## Abstract

Silicon semiconductor-insulator-semiconductor (SIS) structures with high-k dielectrics are a promising new material for photonic and CMOS integrations. The “diode-like” currents through the symmetric atomic layer deposited (ALD) HfO_2_/Al_2_O_3_/HfO_2_… nanolayers with a highest rectification coefficient 10^3^ are observed and explained by the asymmetry of the upper and lower heterointerfaces formed by bonding and ALD processes. As a result, different spatial charge regions (SCRs) are formed on both insulator sides. The lowest leakages are observed through the stacks, with total Al_2_O_3_ thickness values of 8–10 nm, which also provide a diffusive barrier for hydrogen. The dominant mechanism of electron transport through the built-in insulator at the weak field E < 1 MV/cm is thermionic emission. The Poole-Frenkel (PF) mechanism of emission from traps dominates at larger E values. The charge carriers mobility 100–120 cm^2^/(V s) and interface states (IFS) density 1.2 × 10^11^ cm^−2^ are obtained for the n-p SIS structures with insulator HfO_2_:Al_2_O_3_ (10:1) after rapid thermal annealing (RTA) at 800 °C. The drain current hysteresis of pseudo-metal-oxide-semiconductor field effect transistor (MOSFET) with the memory window 1.2–1.3 V at the gate voltage |V_g_| < ±2.5 V is maintained in the RTA treatment at T = 800–900 °C for these transistors.

## 1. Introduction

Recently, the high energy efficiency of optical switches based on semiconductor-insulator-semiconductor (SIS) structures has been reported, reaching femto- and even attojoules per operation when using a design of hybrid structures based on III-V/Si semiconductor pairs [1,2,3,4,5]. The development of more energy-efficient switches, compared to thermo-optical or MEMS devices, means real progress in the implementation of not only fast intrachip communication with switch frequencies of up to 200 GHz [4,5,6,7,8], but also perspectives in building deep learning artificial neural networks based on ferroelectric transistors (FeFET), optical or quantum gates [9,10] compatible with the industrial CMOS technology. Hybrid III-V/Si structures noticeably increase the complexity of mass production of integral circuits (ICs) by the industrial CMOS silicon technology. Thus, the development of all silicon-based switches is a task of current interest.

One of the options for reducing energy consumption in the CMOS technology is the embedding of ferroelectric materials, which provide the inherent amplification of potentials at semiconductor-ferroelectric (FE) interfaces and reduce losses in FeFET keys. Model calculations have shown that using FE insulators provides the possibility of switching for optical phase shifters in the GHz frequency range with a rather low phase shift V_π_L~0.1 V cm at bias voltages of V~1 V on a SIS silicon-based structure with a 20 nm FE Y:HfO_2_ nanolayer [11]. The parameter of FE-losses ρ must not exceed ρ ≅ 10 Ohm∙cm for these semiconductor–ferroelectric–semiconductor (SFS) structures. The real fabrication of similar structures is possible using direct bonding techniques, but it encounters certain difficulties—the low temperature stability of amorphous FE nanolayers, formation of gas blisters, crystallites, and high interface state density D_it_ at the bonded boundary—that reduce the charge carrier mobility in a silicon layer, leading to high leakage currents through the insulator [1,12]. The latter challenge can be solved by using a stable wide bandgap amorphous dielectric or a reverse bias for semiconductors with different types of conductivity instead of a direct bias in SIS structures, as has been recently shown [13]. In that case, high charge carrier mobility and low D_it_ values could be obtained at the interfaces [1].

The other important issues are the lowering of the SIS diode bias voltage and the energy consumption for using these devices in the Internet-of-Things. The well investigated Schottky barrier (SB) FE diodes, where a fully symmetrical metal–ferroelectric–metal (MFM) structure works as a rectifying diode due to the lowering of SB at one MF interface and its increase at the other one due to polarization fields [14,15,16,17,18,19,20,21,22], are an interesting prospect. However, we did not find in the literature any reports devoted to forming similar FE diodes using homojunction silicon layers in SFS structures. These could provide low-voltage FE diode operation if the well-known dependence of ferroelectric properties on interface boundary conditions did not suppress the polar phase formation [23].

The aim of this work was the investigation of the electrical properties of both SIS and SFS silicon structures formed by direct bonding and the implanted hydrogen transfer of a silicon layer onto a Si substrate with an FE stack based on HfO_2_/Al_2_O_3_ nanolaminates. Inserting highly thermally stable Al_2_O_3_ layers increases the thermal stability of the ferroelectric properties of SFS structures during subsequent treatments at the CMOS front-end-of-line (FEOL) fabrication processes instead of the low thermal RTA treatment at 450 °C for 50 s [24]. The quality of these robust SFS structures with composite ferroelectric layers were characterized by their charge carrier mobility and the density of traps at the interphase boundaries, as well as by determining the current leakage mechanisms in the charge carriers enrichment modes (“direct” branch of the I-V curve), and the depletion mode (“reverse” branch) in the space charge region (SCR).

The outline of this paper is as follows. In Section 2, we explain our SIS and SFS wafer specifications and the fabrication process. The electric characteristic measurements are also briefly reviewed. In Section 3, we discuss our approaches to controlling the leakage and rectification currents in SIS structures using symmetrical or asymmetrical stacks of two different alumina and hafnia (Al_2_O_3_ and HfO_2_) dielectric nanolayers changing their order in stacks but not the full thickness. In Section 4, we briefly discuss the leakage current dominant mechanisms and the hysteresis of transfer characteristics of our SFS pseudo-MOSFET (metal-oxide-semiconductor field effect transistor) structures in more details, because these new structures could potentially find widespread use in artificial intelligence integrated circuits (AI IC) due to their high-quality electronic transport and memory properties. In Section 5, we conclude and discuss the future outlook.

## 2. Materials and Methods

Thin films of stacks and nanolaminates of alumina/hafnia (Al_2_O_3_/HfO_2_), as well as single Al_2_O_3_ or HfO_2_ layers, were grown as an insulator for SIS structures by plasma-enhanced atomic layer deposition (PEALD) on (001) silicon wafers. Atomic layer deposition was carried out in the FlexAl tool (OIPT, Yatton, Bristol, UK) using a 13.56 MHz inductively coupled (ICP) plasma source. The SiO_x_N_y_ interlayer between silicon and high-k layer was formed by the plasma nitridization of residual native silicon oxide to reduce the density of electrically active defects on the surface. The nitridization at T = 500 °C, N_2_ pressure *p* = 20 mTorr and ICP discharge power W = 400 W during 180 s were provided in situ immediately before the ALD process at 250–300 °C in the same chamber. In the alumina deposition process, the trimethylaluminum (TMA) precursor was used as an organometallic precursor, and oxygen plasma as an oxidizing agent. The Tetrakis(ethylmethylamido)hafnium(IV) (TEMAH) precursor and also oxygen plasma were used to grow the hafnia film.

SIS structures were formed by the direct bonding technique and hydrogen transfer by the H_2_^+^ ions (E = 120 keV) implantation into the Si donor wafer, then by the bonding with Si n- or p-type substrates and by the thermally induced cleavage by SmartCut^®^ or DeleCut methods (See Appendix A, Appendix A). As a result, silicon and insulator layers simultaneously, or only a silicon layer, were transferred on the substrates, respectively [25]. Before the vacuum bonding of the surface of a pair of silicon wafers, they were processed in Radio Frequency (RF) plasma of nitrogen and oxygen for 60 s. The transfer of n-Si layers from (001) 4.5 Ohm·cm (3–6 × 10^14^ cm^−3^) wafers implanted with hydrogen to similar or p-Si (001) 10 Ohm·cm (1–2 × 10^15^ cm^−3^) substrates with a precoated 20 nm stack or nanolaminates HfO_2_/Al_2_O_3_, or single 20 nm layers of HfO_2_ and Al_2_O_3_, was carried out by the DeleCut method using the vacuum cleaning and temperature bonding (VCTB) in vacuum at a temperature of ~100 °C to increase the effective bonding area and to suppress the large stress generation during the followed thermal splitting of hydrogen implanted Si wafer (Figure 1) [25]. Using the implanted hydrogen transfer of only Si layers makes it possible to avoid the defect generation in the high-k stack at the H implantation and subsequent thermal treatments.

The same last procedure was used for the SmartCut^®^ transfer of both silicon and high-k stack layers on the Si substrate instead of the atmospheric bonding and cleavage (Appendix A, Appendix A). The thermal cleavage (T = 450° C) resulted in the transfer of the layer package of Si/high-k stack on the Si-substrate, or only a Si-layer onto the high-k stack/Si-substrate. These SIS wafers, similar to usual silicon-on-insulator (SOI), with a high-k dielectric instead of silica, were subjected to a sequential furnace or rapid thermal annealing in an atmosphere of argon or nitrogen at temperatures of 650–1100 ° C (furnace annealing and rapid thermal annealing). With respect to the samples, the Si films in the SIS samples were thinned by the sequential operations of chlorine oxidation at the 1100 °C/wet etching (O/E) of the oxidized layer in a 1% HF solution to the Si layer thicknesses of 50–300 nm.

The C-V and I-V measurements and drain-gate characteristics of pseudo-MOS transistors were used on the lithographically created mesa structures in the upper Si layer sized 1 × 1 mm^2^ or in the Corbino geometry in the transferred Si layer of 1 × 1 mm^2^ in size. Al contacts in the Corbino geometry after the post-metallization annealing had a 0.6 mm inner circle, and electrodes with an outer ring diameter of 3 mm on the transferred silicon layer. The Si layer outside the lithographic masks was removed before measurements in boiling ammonia at 200 °C. The measurements were carried out with the Keythley 4200 unit (Tektronix Inc. of Beaverton, OR, USA) using tungsten probe needles at the distance of 100 µm and the tip radius of 20 µm with the clamping force of 60 g as drain and source contacts, and with the back gate, which was the Si-substrate with the ohmic InGa contact. The layer thicknesses of SIS structures were controlled by spectral ellipsometry (SE) after each step of sample fabrication.

## 3. Results

### 3.1. Symmetrical SIS with a 20-nm-Thick Alumina or Hafnia Built-In Insulator

The data in Figure 2, Figure 3, Figure 4, Figure 5 and Figure 6 present only the last isochronal temperature data, where the slow interface states were essentially removed. These slow interface states provide huge (3–15 V) unstable C-V and I-V hysteresis on these defect states after the thermal treatments in the range 600–800 °C similar to the flash memory that out the scope of interest. Their removal was controlled by C-V and pseudo-MOSFET measurements at different sweeping rates (see e.g., Figure 1, Figure 4 and Figure 8). To minimize the leakage currents through the SIS structure, a 20-nm-thick alumina layer was firstly investigated as an insulator after different thermal treatments due to its high bandgap and amorphous phase stability [26,27,28].

Figure 2a shows the hysteresis-like drain-gate characteristics of n-SIS pseudo-MOSFETs with a 500 nm n-Si layer and a 20 nm insulator layer of an ALD Al_2_O_3_ layer immediately after the furnace annealing at 450 °C for the sweep range of V_g_ = −1.0 + 0.5 Volts. The gate voltage V_g_ was applied to the n-Si substrate, and the sweep rate was varied from 50 V/s to 3.3 × 10^−3^ V/s. A decrease in the current value with a decrease in the negative bias indicates a hole conductivity and a built-in negative charge in the insulator. The observed hysteresis behavior of I-V is apparently due to the slow interface states (IFS) recharging in the insulator. The charge capture on those levels has a typical dependence on the sweep speed (Figure 2a, dashed arrows).

A further FA or RTA at the increased temperatures 950–1000 °C does not lead to a slow IFS removal according to the C-V measurements at different frequencies (Figure 2b) or crystallization of amorphous ALD Al_2_O_3_, but eliminates the residual defects in the transferred silicon film and also leads to the formation of silicon oxide interlayers in the insulator boundaries and to the formation of positively charged oxygen vacancies in aluminum oxide [26,27]. As a result, a large positive charge appears in the Al_2_O_3_ insulator, and the field effect is almost impossible to observe due to the electron accumulation at the insulator boundaries.

Huge changes in the drain current in n-SIS pseudo-MOSFET structures occur when the gate voltage V_g_ changes in the interval from −8 to +8 Volts, even after rapid thermal annealing (RTA), due to leaks through the insulator layer (Figure 3 and Figure 4). The leakage currents were measured with an alternately grounded source or drain contacts. They decrease at bias voltages |V_g_|~1 V, with respect to V_g_ around 0 V, and demonstrate pronounced “diode like” characteristics at |V_g_| > 2 V. The curves could also be reversed when the location of the probes or mesa structures on the wafer changed. Moreover, the changes in the ratio of leakage currents through the insulator from 1 to 10 were not proportional to the ratio of the source and drain Al contact areas (A_s_ = 3.9 × 10^−2^ cm^2^, source contact with Probe 1 and A_d_ = 3.0 × 10^−3^ cm^2^, drain contact with Probe 2) for the mesa structures with Corbino geometry, which is about 1 in Figure 3. Using these contact areas and 0.6 mm distance between them we can estimate the defect density.

This behavior indicates the presence of randomly located defect critical density N_R_ in the insulator layer with a value of N_R_ = 1/A_d_ < 3.3 × 10^2^ cm^−2^, which determines the current flow through the buried dielectric, similar to the phenomena in oxide-based memristors. At voltages |V_g_| > 1 V, the currents and noises at I-V curves sharply decrease due to the carrier depletion effect in the silicon layers adjacent to the insulator in the SCR on one of the sides, and then the accumulation of electrons or holes on both sides of the dielectric boundaries, and vice versa, when the polarity changes (Figure 3). Probably, accumulation and inversion phenomena can switch currents through the defects in the insulator layer, and they are responsible for the charge carriers transfer if they propagate to one of the insulator boundaries. These defects are similar to those previously observed by TEM grain boundaries [25,27], or lonely conducting filaments and behave like ohmic shunts; the current through these defects increases linearly with V_g_ and does not depend on the contact area. It is possible that the conductive filaments in the insulator are a consequence of electroforming processes, the same as the formation of memristor cells. More likely, such defects begin from the boundary of the bonding interface, and that corresponds, in most cases, to leaks with a positive bias voltage V_g_.

The buried insulator with amorphous HfO_2_ layers crystallizes at much lower temperatures T~600–800 °C [16]. The hafnium dioxide band gap is also smaller than that of alumina, and it leads to an increase in the leakage currents in both furnace and rapid thermal annealing (RTA) (Figure 4). Reducing the leakages ensures the introduction of Al_2_O_3_ layers into the insulator based on hafnium dioxide. An increase in the thickness of aluminum oxide layer inserts by a factor of 5 reduces the leakage current by 4 orders of magnitude. However, in the case of Al_2_O_3_ contact with the silicon interface, the problem of built-in positive charge in the insulator arises again and, possibly, that of the formation of dipoles at Al_2_O_3_ or HfO_2_ heterointerfaces with SiO_2_ interlayers [28,29].

### 3.2. Rectification and Hysteresis in the Symmetrical SIS with a 20 nm Stack HfO_2_/Al_2_O_3_/…/HfO_2_

The positive charge in the insulator still exists in the case of a symmetric arrangement of the Al_2_O_3_ insert inside the insulator as HfO_2_/Al_2_O_3_/HfO_2_, but oppositely directed dipoles compensate each other at the heterointerfaces (Figure 5a and Figure 6a) [29]. The probability of dipole recharging can compensate the ferroelectric type hysteresis at the asymmetric n-SIS pseudo-MOSFET’s drain-gate characteristics, but not for the symmetrical ones (Figure 6 and Figure 7). These dipoles determine that “forward” and “reverse diode” characteristics of the leakage currents through the asymmetrical stack depend on their orientations (See Appendix A, Appendix A). An increase in the total stack thickness to 30 nm decreases the gate leakage current I_g_ by more than 2 orders of magnitude (see Appendix A, Appendix A). At the same time, an increase in the furnace annealing temperature to 1000 °C does not lead to the essential increase in the gate leakage current I_g_ due to the thick alumina insert in the high-k layer (Figure 6), but drastically decreases the IFS density and the hysteresis (Figure 5b). The Al_2_O_3_ insert inside the insulator demonstrates a low leakage current at voltages |V_g_| > 1 V even after the furnace annealing at 1000 °C, but without any hysteresis and dependence on the contact areas (Figure 5b and Figure 7a). These defects still show the current rectification due to their different nature of both interfaces.

To clarify the role of the near interface defects, the same structure was investigated with the special defect region in the n-Si substrate introduced by CO^+^ ion implantation before the PEALD deposition and bonding (Figure 6b). Besides an increase in the leakage current, the polarity of the I-V branches was reversed by these defects near the bottom interface showing their role in the diode-like behavior.

## 4. Discussion

### 4.1. Random Shunts in the Built-in Insulator HfO_2_/Al_2_O_3_ Stacks

The absence of I_g_ dependence on the contact area ratio ~13 indicates the predominant contribution of “random” defects in oxide layer leakages after the annealing. It is important to note that this independence is observed for “reverse” current branches and that confirms their relation to the defects not only in the insulator, but also in the SCR. To verify the assumption of a connection of the diode-like behavior with the defects in the SCR, the electron distribution under the insulator after the RTA at 800 °C was measured by the C-V method [29]. The high-resistance p- (i-) layer spreads the SCR in the substrate up to 2 µm and ensures the constant I_g_ leakage current value when the bias voltage V_g_ changed in the interval of +1–+8 V (Figure 7b). It can be seen that the contribution of nonlinear effects becomes dominating at the fields E > 1 MV/cm that can be seen from the ratio of currents at lower voltages (Figure 4).

### 4.2. Possible Mechanisms of the Charge Transport through Built-In Insulator HfO_2_/Al_2_O_3_ Stacks

The I_g_ currents in the forward direction through a thin insulator layer in the case of their significant (by 1–3 orders of magnitude) excess over the backward currents can be determined by a combination of several mechanisms at once: the resistive current through filaments/shunts in the dielectric, thermionic emission (TE) of charge carriers, direct interband tunneling, trap-assisted tunneling (TAT), emission from traps using the Poole-Frenkel (PF) mechanism, and the injection tunneling by Fowler-Nordheim (FN) mechanism. The relative contributions of these mechanisms depend on the layer thickness and electric field strength in the dielectric, as well as its temperature and defects, and can be extracted in various ways from the analysis of field and temperature dependences [30].

The space charge region in the SIS structures with the CO^+^ implantation reduces the field in the dielectric layer due to the applied voltage redistribution between the depletion region and the built-in insulator of the structure, and that does not allow us to draw an unambiguous conclusion about the prevailing conduction mechanisms. Nevertheless, an estimate of the contribution of “random” defects to the leakage current can be obtained from a comparison of the I-V dependences for two different contacts on the same mesastructure. For instance, the leakage currents for Corbino n-type mesa structures on the 500 nm n-Si/Al_2_O_3_ 2nm/HfO_2_ 20nm/Si substrate through the insulator at the positive bias of V_g_> 0.1 V (Appendix A) are caused after the RTA at 950 °C by the electron conductivity from the silicon film to the Si substrate through the oxide layer. The equivalent oxide thickness (EOT) of such BOX layer is about 5 nm in terms of SiO_2_. The currents in the forward direction exceed the currents in the backward direction by three orders of magnitude.

### 4.3. Normalized Differential Conductance Approach for the Separation of Charge Transport Mechanisms

Naturally, the analysis of normalized differential conductance (NDC) gives a better approximation of the linear dependence for high voltages (Figure 7a) for a “direct” drain current with a smaller area and, accordingly, with a lower probability of the contribution of random defects in two coordinates of the abscissa axis 1/V and V^1/2^ [30]. The intersection of the linear approximation with the ordinate axis at NDC = 0.9 corresponds to the Poole-Frenkel mechanism (NDC = 1) for the electron transport through the built-in insulator (See Appendix A, Appendix A) [30,31]. The value NDC = 0.16 for the *x*-axis in the V^1/2^ coordinates has an overestimated NDC value instead of the expected value NDC = 0, which corresponds to the thermionic emission of electrons at lower voltages V_g_ (Figure 7b). A large error for lower V_g_ values corresponds to a smaller fraction of nonlinear effects in the leakage current at low voltages. However, from the data in Figure 7, it follows that the dominant mechanism of the leakage current through the insulator layer from the high-k Al_2_O_3_ (2 nm)/HfO_2_ (20 nm) stack without random defects is the thermionic emission at 0.15 < V_g_ < 0.9 V and the emission from traps by the Poole-Frenkel (PF) mechanism at 1.5 < V_g_ < 4.0 V. The high leakage current in the forward direction at biasing V_g_ > 1 V show linear dependences with the slope *n* = 2 in the log-log scale that corresponds to the Fouler-Nordheim tunneling mechanism in the high electric field E~1 MV/cm (See Appendix A, Appendix A) [32].

The symmetrical n-SIS structures with the FE stack in the insulator can show diode like characteristics due to the insulator polarization at the coercive field E_C_ (Figure 8). The observed asymmetry of the I-V characteristics of leakage currents through high-k dielectric stacks with an equivalent oxide thickness EOT~3–5 nm, even in the case of symmetrical structures of n-Si film_/_Al_2_O_3/_n-Si substrate, or n-Si film/HfO_2_/n-Si substrate, or n-Si film/HfO_2_/Al_2_O_3_/HfO_2_/n-Si substrate could be explained by the unequal distribution of traps at the Si film/insulator and insulator/Si substrate hetero-interfaces, which differ in their formation (bonding and ALD deposition) technologies, as well as in terms of different concentrations of hydrogen atoms at the upper (bonding and hydrogen transfer processes) and lower (ALD deposition) heterointerfaces measured by the SIMS technique (Appendix A, Appendix A). Hydrogen is involved in the oxygen vacancies formation and promotes the oxygen movement from the insulator layers to the boundaries with silicon. As a result, a thicker silicon oxide interlayer and a higher oxygen vacancies concentration in the insulator are formed at the upper boundary. The higher hydrogen content at the upper boundary corresponds to a high probability of detecting both random shunt defects with ohmic characteristics and direct quasi-diode characteristics with a rectification coefficient of up to 10^2^ at positive bias voltages on the substrate. The asymmetric structures of the n-Si film/HfO_2_/Al_2_O_3_/n-Si substrate and the n-Si film/Al_2_O_3_/HfO_2_/n-Si substrate demonstrate a 1–2 orders of magnitude higher rectification coefficient. At the same time, significantly higher I_ds_ currents are observed at V_g_~0 Volt of the drain-gate characteristics of n-SIS pseudo-MOSFETs due to the large positive charge inside the insulator near the n-Si/Al_2_O_3_ heterointerfaces, which include the SiO_2_ interlayer, and a lower slope of characteristics due to the SCR additionally formed in the substrate by the CO^+^ ion implantation (Figure 7). An SCR expansion using a CO^+^ implanted getter also increases the rectification coefficient of these structures.

### 4.4. Ferroelectricity in Nanolaminated Built-In Insulator HfO_2_/Al_2_O_3_/HfO_2_ Stacks

The symmetrical SFS structures provide electrically switchable rectification due to the inherent switchable electric field. The band bending asymmetry, relative to the polarity of bias voltage, is demonstrated in Figure 8. This polarity depends on the up or down polarization even in the case of highly symmetric SIS structure. Moreover, the high remnant polarization P_r_ provides a simultaneously strong accumulation and inversion at both interfaces in the slightly doped or undoped semiconductors. In other words, the ferroelectric polarization charges ± Qr form the space charge region similar to the one in the p-n junction if their charges are higher than those of trapped ones |Q_r_| > Q_f_.

The asymmetrical structures of the n-Si film_/_HfO_2_:Al_2_O_3_ (10:1)/p-Si substrate (np-SFS) demonstrate, in addition to good insulating properties I_g_ < 10^−7^ A/cm^2^ at |V_g_| < 4 V, a significant ferroelectric (FE) hysteresis, with a ΔV_g_~1.0–1.3 V memory window (MW) of the drain-gate characteristics of np-SIS pseudo-MOSFETs, as well as a saturated drain current *I_d,s_* (Figure 9). This current is saturated due to a long pseudo-MOSFET channel with Schottky barrier contacts [33,34]. The voltage memory window ΔV_g_ is expressed through the FE remnant polarization P_r_ as [16]:ΔV_g_ = V_Tp−_ − V_Tp+_ = 2P_r_ε_0_ε_HAO_/δ,(1)
where = V_Tp−_ − V_Tp+_ is the difference of the float band voltages for two polarizations and that coincides with the hole threshold voltage difference (see Appendix A, Appendix A), *δ* is the distance between the charges in the FE insulator and in the channel ~1 nm. Equation (1) gives the value *2*P_r_ = 23 µC/cm^2^, which corresponds to the published results for structures fabricated under similar conditions [35].

The C-V, G-V and P-V loops, as well as the PUND pulse I-t measurements and PFM hysteresis, presented by Appendix A (Appendix A) after the further RTA treatment at 950 °C confirm the ferroelectric nature of the transfer characteristic hysteresis in Figure 9. The last RTA treatment decreased the memory windows 3.7 times and that corresponds to the estimated value 2P_r_~6 µC/cm^2^ (Appendix A). Triangular pulses were applied to get the P-E hysteresis. The nonsymmetrical P-V loops with gaps are the result of reverse biasing in n-p SFS mesa structures. The PFM-V hysteresis is more symmetrical after n-Si layer removal (Appendix A).

### 4.5. SFS Pseudo-MOSFET Characteristics

The maximum mobility values for electrons and holes of SIS structures with the ferroelectric hysteresis were determined for pseudo-MOSFETs by the Y-function method according to the following formula [36]:μ_n,p_ = (β_n,p_)^2^/(f_n_C_BOX_V_DS_),(2)
where β_n,p_ are the slopes of Y-function branches, f_n_ = f_p_ = 0.75 or 5.3 is the geometric factor for two-probe measurements, or in Corbino geometry, respectively, C_BOX_ is the buried oxide (BOX) capacity C_BOX_ = ε_0_·ε_BOX_/t_BOX_ (dielectric constant *ε*_BOX_~20), V_DS_ = 0.1 V is the drain voltage.

The silicon layer mobility values µ, determined from the maximum slopes of the linear branches of Y-functions (Figure 9), were µ_n_ = 120 cm^2^/(V s) and µ_p_ = 110 cm^2^/(V s) for electrons and holes, respectively. The density of states D_it_ were calculated using Equation (3) and the data (Appendix A, Appendix A), where there are only two linear segments on the Y-function branches:V_Tn_ − V_Tp_ ≅ 2·Φ_F_ + (q·t_BOX_/ε**_0_**·ε_BOX_)·(N_0_·t_Si_ + 2·Φ_F_·D_it_),(3)
where q is the electron charge, N_0_ is the donor concentration, Φ_F_ = E_F_ − E_i_ = 0.144 eV is the Fermi level position in the silicon layer bulk with donor concentration N_0_ = 4 10^14^ cm^−3^, D_it_ is the density of states at the upper heterointerface and t_Si_ is the top silicon layer thickness.

Then, according to the value of V_Tn_ − V_Tp_ on the thickness of t_BOX_ (Figure 9 and Appendix A) and Equation (2), the value of D_it_ = 1.2 10^11^ cm^−2^ for the SFS with a 20 nm high-k stack of layered nanolaminate HfO_2_:Al_2_O_3_ (10:1). From these data, it can be concluded that the RTA treatments at 800 °C of the bonded silicon layer on the substrate with the hafnia-alumina-based nanolaminate built-in insulator provide the fabrication of Silicon–Ferroelectric–Silicon wafers with charge carrier mobility µ_n_,_p_ in the Si layer at levels of µ_n_ = 120 cm^2^/(V s) and µ_p_ = 110 cm^2^/(V s) for electrons and holes, respectively (Figure 6a and Figure 9). Keeping the carrier mobility and FE hysteresis after the rapid thermal annealing at T~800 °C is required for optical SIS phase shifters, FeFET embedded nonvolatile memories and Silicon-based Associative Neural Network technologies compatible with the industrial CMOS approach [10,13,37].

## 5. Conclusions

Investigations into the electrical characteristics of pseudo-MOSFETs mesa structures in n-SOI wafers were carried out with a 20 nm insulator with different high-k dielectrics layers—symmetric and asymmetrical stacks of the Al_2_O_3_/HfO_2_, and nanolaminate of HfO_2_:Al_2_O_3_—both in the two-probe measurement geometry and in Corbino geometry. The quasi-diode character of leakage currents through buried high-k built-in dielectrics with an equivalent oxide thickness EOT = 3–5 nm was detected, even in the case of symmetric semiconductor–insulator–semiconductor (SIS) structures with n-Si film/HfO_2_/Al_2_O_3_/HfO_2_/n-Si substrate. The reason for the diode-like characteristics in symmetrical SIS structures is explained by the asymmetry of the upper and lower heterointerfaces formed by the silicon bonding to the high-k insulating layer and ALD deposition of that layer on a silicon substrate, respectively. An additional factor is the different concentration of residual hydrogen at these interfaces, and it segregates in a subsequent SOI wafer annealing at T = 800–1000 °C. As a result, different spatial charge regions (SCRs) are formed near both Si/insulator interfaces.

The smallest leakage currents were obtained with high-k stacks containing an 8–10 nm Al_2_O_3_ layer, which also provided a barrier to the diffusion penetration of residual hydrogen into the substrate. In the SIS structures with the Al_2_O_3_ 2 nm/HfO_2_ 20 nm stack and a rectification coefficient of 10^3^, it was shown that the dominant mechanism of electron transport through the dielectric at weak fields E < 1 MV/cm is thermionic emission, and at large E values, the Poole-Frenkel (PF) mechanism of electron emission from traps in the insulator dominates.

The maximum carrier mobility μ = 100–120 cm^2^/(V s) and the minimum density of states D_it_ = 1.2 10^11^ cm^−2^ were obtained for the asymmetrical np-SIS structures with a high-k insulating stack of nanolaminate HfO_2_:Al_2_O_3_ (10:1) after the RTA at 800 °C. The FE hysteresis still remains with a memory window of MW = 1.0–1.3 Volts at 0 < V_g_ < 3 V after the RTA processing of these SIS structures, and that makes it possible to use them as SFS substrates in the industrial CMOS process.

Silicon SIS structures with high-k stacks and nanolaminates of hafnia-alumina and bonded silicon layers of n- and p-type conductivities are promising for optical phase shifters at reverse biases, as well as for their use in low-energy ICs, FeFET embedded nonvolatile memories and the Associative Neural Network circuitry due to the compatibility of such approach with the industrial CMOS technology and the preservation of the ferroelectric effect at the front-end-of-line CMOS processes, as we have shown it for the ferroelectric stack of HfO_2_:Al_2_O_3_ (10:1). The properties of the nonvolatile memory cells with FeFETs on the SFS substrates will be the goal of future research directions.

## Figures and Tables

**Figure 1 nanomaterials-11-00291-f001:**
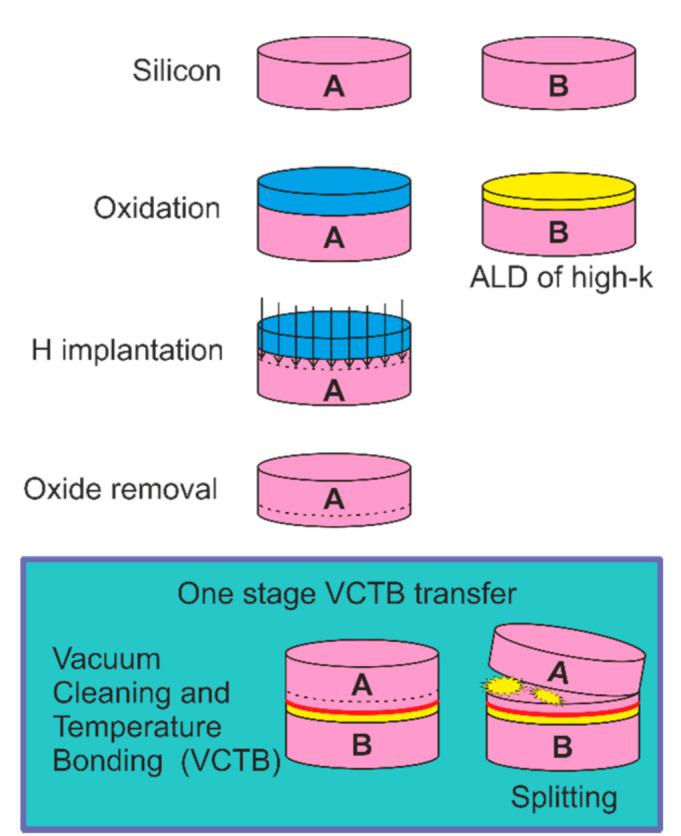
The SFS (semiconductor–ferroelectric–semiconductor) structure process flow (for more data see the [25,26]).

**Figure 2 nanomaterials-11-00291-f002:**
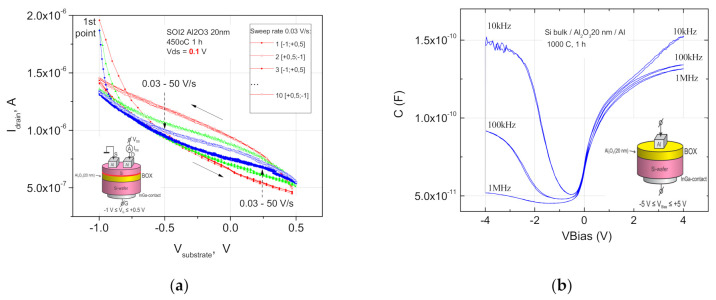
(**a**) Hysteresis-like behavior of the drain-gate characteristics at the range V_g_ = −1.0 + 0.5 V of n-SIS pseudo-MOSFET (metal-oxide-semiconductor field effect transistor) with the 500 nm n-Si/20 nm Al_2_O_3_/n-Si substrate after the furnace annealing (FA) at 450 °C during 1 h. Potential V_g_ was applied to the n-Si substrate. The drain voltage V_DS_ was 0.1 V for all pseudo-MOSFET measurements. The sweep rate was 50 V/s, 5.0 × 10^−1^ V/s, 3.3 × 10^−3^ V/s (shown by dashed arrows). In the inset are the measurement scheme and the layers of the n-SIS structure, where the bonding interface is indicated by a red line; (**b**) C-V plots of the MIS structure with high-k insulator Al_2_O_3_ after the final FA at 1000 °C for 1 h and top Si layer etching. In the inset are the measurement scheme and the n-type MIS structure layers.

**Figure 3 nanomaterials-11-00291-f003:**
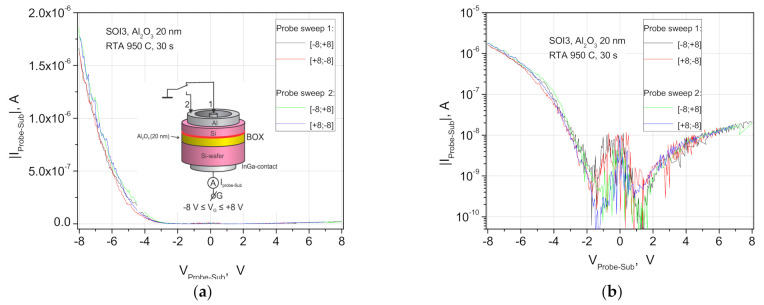
(**a**) Diode-like I-V characteristics of the source current-gate voltage (Probe sweep 2) and drain current-gate voltage (Probe sweep 1) contacts of n-SIS pseudo-MOSFETs on the structures 500 nm n-Si/20 nm Al_2_O_3_/Si-substrate after the RTA treatment at 950 °C for 30 s. The I-V curves are shown for the sweep in the range V_g_ = −8 + 8 V on the linear scale; (**b**) the same on the semi-logarithmic scales.

**Figure 4 nanomaterials-11-00291-f004:**
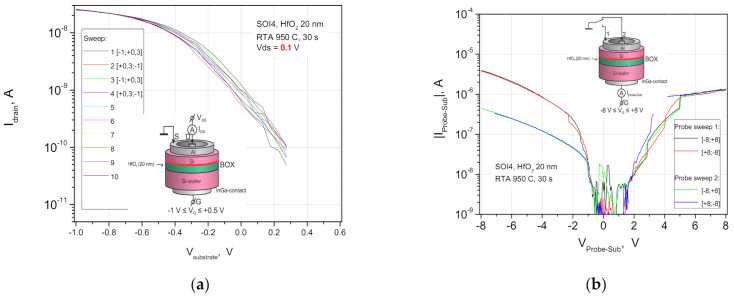
(**a**) Drain-current-gate voltage characteristics of the n-SIS (500 nm n-Si/20 nm HfO_2_/Si substrate) pseudo-MOSFETs after the RTA at 950 °C for a scan of V_g_ = −1.0 + 0.6 V; (**b**) current-voltage characteristics at the source-gate and drain-gate contacts, bias voltages in the range V_g_ = −8 + 8 V.

**Figure 5 nanomaterials-11-00291-f005:**
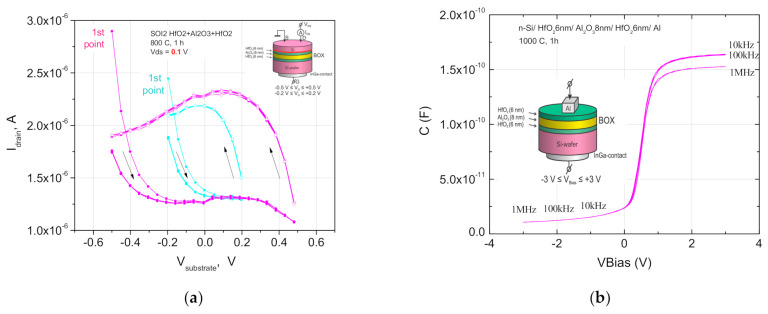
(**a**) Hysteresis behavior of the electron current in the transfer characteristics of n-SIS pseudo-MOSFETs with the 500 nm n-Si/20 nm HfO_2_ 6 nm/Al_2_O_3_ 8 nm/HfO_2_ 6 nm/n-Si substrate on the mesa structures with tungsten needle-shaped contacts (see inset) after the FA at 800 °C for 1 h. Constant sweep rate 50 V/s in the two ranges of V_g_ = −0.2 + 0.2 V (cyan) and V_g_ = −0.5 + 0.5 V (magenta) is set for measurements with the starting (1st) point at V_g_ = −0.2 or −0.8 V during multicycle measurements. The gate voltage V_g_ is applied to the n-Si substrate; (**b**) C-V plots of the MIS structure with high-k insulator stack HfO2/Al2O3/HfO2 after the FA at 1000 °C for 1 h and top Si layer etching. In the inset are the measurement scheme and the n-type MIS structure layers.

**Figure 6 nanomaterials-11-00291-f006:**
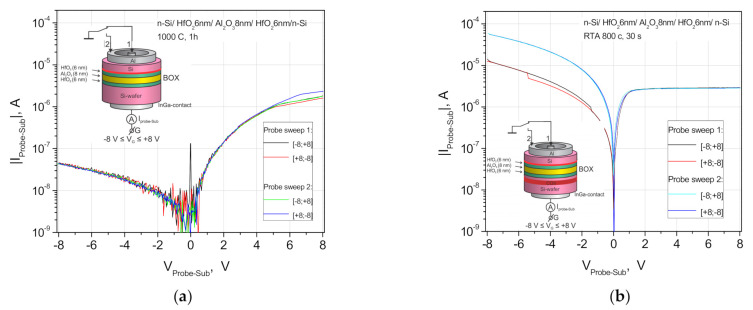
(**a**) Diode-like current-voltage (I-V) characteristics at the source–gate contacts and drain-gate contacts of n-SIS Corbino structures on the wafers fabricated by the DeleCut method: 500 nm n-Si/HfO_2_ 6 nm_/_Al_2_O_3_ 8 nm/HfO_2_ 6 nm_/_n-Si-substrate, after the FA at 1000 ° C for 1 h; (**b**) the same, but with a CO^+^ implanted getter (E_CO+_ = 90 keV, Ф = 1 × 10^16^ cm^−2^) after the RTA at 800 °C for 30 s. The bias voltage is applied to the n-Si substrate. In the insets are the measurement scheme and the n-SIS structure layers, where the bonding interface and COII getter are indicated by a red line and stripe, respectively.

**Figure 7 nanomaterials-11-00291-f007:**
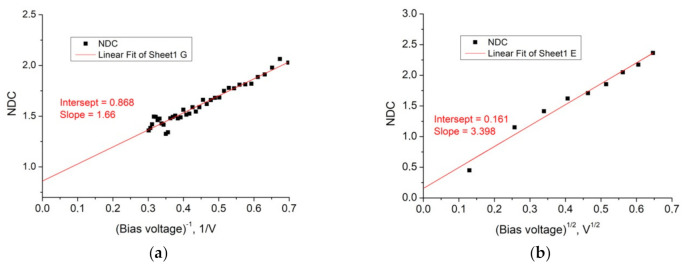
(**a**) Linear approximations of normalized differential conductance (NDC, according to [28], for the leakage current at the drain–gate contacts in the “forward” direction shown in the coordinates of 1/V for a larger bias voltage based on the data extracted from Appendix A; (**b**) the same in the V^1/2^ coordinates for a lower value of bias voltage.

**Figure 8 nanomaterials-11-00291-f008:**
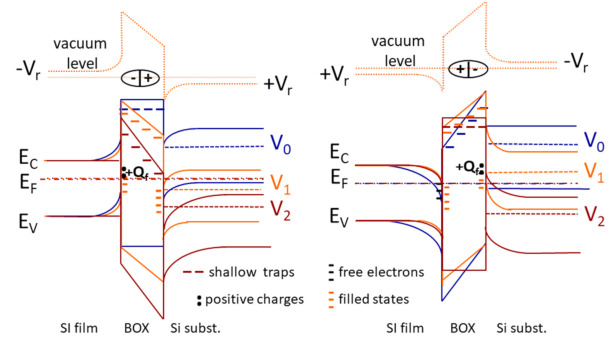
Silicon and hafnia bandgaps (not on the same energy scales for picture clarity) bending at the ferroelectric switching in the symmetric n-SIS structure with the two polarizations down to the substrates (left) and to the Si film (right), the shallow and deep electron traps with the +Q_f_ net trapped charges at the boundaries of ferroelectric stack HfO_2_:Al_2_O_3_ (10:1) for the different Si substrate bias voltage V_sub_: V_0_ = -V_R_, where V_R_ is the residual polarization voltage after/before the P-down switching (left), V_1_ = 0 V, V_2_ = V_R_, where V_R_ is the voltage for the coercive switching of polarization (left). The same for the P-up switching in the built-in insulator (right). The Si film is grounded. The vacuum levels are shown for the V_1_ voltages only.

**Figure 9 nanomaterials-11-00291-f009:**
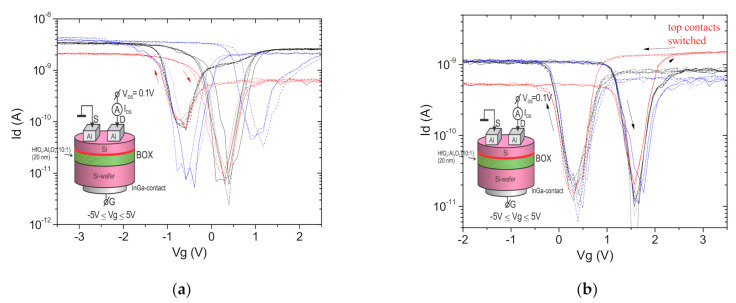
Ferroelectric hysteresis at the transfer characteristics of pseudo-MOSFET on the np-SIS mesastructure with the 500 nm n-Si/ 20 nm built-in insulator BOX nanolaminate HfO_2_:Al_2_O_3_ (10:1)/n-Si substrate: (**a**) after the RTA at 800 °С and (**b**) after the RTA at 900 °С for three different sweep rates: 0.4 V/s (black), 0.12 V/s (red) and 0.04 V/s (blue) and three sweep cycles (solid, dot and dash-dot lines).

## Data Availability

The data presented in this study are available on request from the corresponding author.

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
