# Peer review of "Diode-Like Current Leakage and Ferroelectric Switching in Silicon SIS Structures with Hafnia-Alumina Nanolaminates"

_nanomaterials, 2021, doi:10.3390/nano11020291_

Round 1
Reviewer 1 Report
Title: Diode-like current leakage and ferroelectric switching in silicon SIS structures with hafnia-alumina nanolaminates
Main point:
- Absence of origin of NR <3x10-3cm-2 in page 5. line 171: It is hard to find that the author how to calculate the density.
- In order to explain the effect of defect in tunneling mechanism, we use some fitting data based on the measured data. However, author didn’t suggest any fitting data or supporting data to explain the tunneling mechanism related to defects in oxide layer and interface.
- The explanation is promiscuous. Specifically, even if author said that ‘the source current – gate voltage (probe sweep2) and drain current- gate voltage (probe sweep1)’ at figure 2(a), the data of y-axis means Iprobe-sub. It is very hard to understand the data because this manuscript is not complete and very messy.
- Many figures had mixture of unorganized and insufficient data, such as various temperature without data.
- Although ferroelectric materials were used in device, it is hard to find out any advantage in device characteristics as well as any suggestion for something new, such as difference between pseudo-MOSFET characteristics at this paper and at another. As you know, ferroelectric properties in Al2O3/HfO2 = 1:9 is already well known.
Overall, the main issue in this manuscript was not well explained as well as experimental data was insufficient to prove the issues.
Reviewer 2 Report
This paper entitled “ Diode-like current leakage and ferroelectric switching in silicon SIS structures with hafnia-alumina nanolaminates” described the MOS diode and MOSFET characteristics fabricated by the bonding process with high-k HfO2/Al2O3 and ferroelectric HfAlO2. The authors revised the paper with following the reviewer’s comments. However, the authors still necessary to revise especially for the several important parts of this paper. Therefore, the reviewer thinks the major revisions are necessary. The comments for the authors are listed below.
1.Experimental procedure should be clearly described in section 2 using schematic cross section, plane view etc. for the readers. Then, the device structure and size for each measurement should be described.
2. “BOX” description is still used in some descriptions.
3. The authors investigated the device with bottom gate structure so that the heavily doped Si should be used for the bottom Si substrate as a gate. The lightly doped substrates are not adequate for the gate operation.
4. The interface for the channel should not be formed by the bonding.
5. In Fig. 1, the authors explained the influence of positive charge in Al2O3. However, it is the well-known property of the Al2O3 gate insulator.
6. In Fig. 1(b), the authors should explain the results of 600oC and 800oC annealing.
7. In Fig. 2, the reviewer did not understand why the authors explained as “diode like” characteristics. Figure 2 shows the diode characteristic.
8. In Fig. 3, the authors should show the data for the RTA of 600oC and 800oC annealing.
9. In Fig. 4(a), the authors should explain what the 1st point is. The difference of line colors should also be explained for the readers.
10. In Fig. 4(b), the authors should show the data for the RTA of 600oC and 800oC annealing.
Reviewer 3 Report
The manuscript reports the study on Silicon semiconductor-insulator-semiconductor (SIS) structures with hafnia-alumina nano-laminates, focusing on the effects of diode-like leakage current and ferroelectric switching. The device structures are based on Si semiconductor, which is feasible for future potential uses in microelectronics and show very good device performance, including charge carriers mobility of 100-120 cm2/(V s), the minimum density of states of 1.2x1011 cm-2, and the FE hysteresis effect in a memory window of 1.0-1.3 V in the gate voltage range of 0 - 3 V. The work is original. The results and analysis is interesting to the researchers in the field.
The manuscript needs some improvements prior to acceptance.
- In line 62 – 64 (page 2), it writes “The aim of this work was the fabrication of both SIS and SFS silicon structures by direct bonding and the implanted hydrogen transfer of a silicon layer onto a Si substrate with an FE stack based on HfO2 / Al2O3 nanolaminates.”
Actually, the aim of the work was not the fabrication of the SIS and SFS structure, rather the studies on diode-like leakage current and ferroelectric switching effects. Although the manuscript reports the device fabrication, it does not present any experimental device structures. Therefore, I suggest to change the way of statement.
- In line 170-171 (Page 5), it writes “defect critical density NR in the insulator layer with a value of NR <3x10-3cm-2”
The NR number is apparently wrong. After putting the right number, the authors need to explain how to get the number experimentally.
- In line 178-180, “These defects are similar to lonely conducting filaments and behave like Ohmic shunts; the current through those defect increases linearly with Vg and does not depend on the contact area.”
Here, suggestively, an SEM or a TEM result can be presented to show experimental evidence to support the statement. It is also suggested that an SEM or a TEM image be used to show the real device structure, to make the work more complete.
- Carefully check the material parameters in the text again. Also, breaking up the long sentences in the text to making them more readable. For examples, those in lines 94 -- 98 and in lines 114 – 119.
Reviewer 4 Report
This work investigated the electrical characteristics of pseudo-MOSFETs mesa structures in n-SOI wafers with a 20 nm insulator of different high-k dielectrics layers: symmetric and asymmetrical stacks of the Al2O3/HfO2, nanolaminate of HfO2:Al2O3 both in the two-probe measurement geometry and in Corbino geometry. This work will be of interest to other researchers in scientific and engineering community of photonic and CMOS integrations. At the same time, the reviewer has to point that in current manuscript there are concerns to be addressed. The concerns are:
1) In introduction, the authors write: “However, hybrid III-V/Si structures noticeably increase the cost of mass production of integral circuits (ICs) by the industrial CMOS silicon technology.” The introduction have room to be further improved. Recently, hybrid III-V/Si structures have been applied in transistor technology and LEDs. It would be great if the authors include these new developments and achievements in the introduction, so to give the readers a much broader view. Several important references related to the hybrid III-V/Si structures, such as IEEE Transactions on Electron Devices 67(12), 5306 - 5314(2020); Nanomaterials 2019, 9, 1178; Optics Express 27, A1506 (2019), etc. should be added, so that the readers can be clear about the state-of-the-art of this topic.
2) In the captions of Figures, “insert” should be revised to be “inset”.
3) The authors should define the full name of RTA when the abbreviation first occur in the text.
4) A schematic illustration of the fabrication process of SIS/SFS structures is encouraged.
5) Can you make a comparison between the novel SIS/SFS structures proposed in this work and conventional hybrid III-V/Si structures.
6) There are some grammatical errors in the manuscript, although most of them do not obscure the understanding of the technical contents. For example:
--“In Figure 1a is the hysteresis-like drain-gate characteristics of n-SIS pseudo-MOSFETs” should be corrected to be “Figure 1a shows the hysteresis-like drain-gate characteristics of n-SIS pseudo-MOSFETs”.
--“The observed hysteresis behavior of I-V is apparently due to the slow interface states (IFS) recharging in the insulator.” should be corrected to be “The observed hysteresis behavior of I-V is apparent due to the slow interface states (IFS) recharging in the insulator.”
Round 2
Reviewer 2 Report
The authors revised some of the parts in this paper. The reviewer thinks further revisions are necessary for the publication as mentioned before. However, some of the results are interesting for the readers in this field. Therefore, the reviewer recommends the authors revise carefully with following the reviewer's comments.
This paper entitled “ Diode-like current leakage and ferroelectric 2 switching in silicon SIS structures with hafnia-alumina nanolaminates” described the MOS diode and MOSFET characteristics fabricated by the bonding process with high-k HfO2/Al2O3 and ferroelectric HfAlO2. However, the explanation of the fabrication process and experimental results are not clear, and some of the discussion and explanation for the obtained result are not correct. Therefore, the reviewer thinks the major revisions are necessary. The comments for the authors are listed below.
- Experimental procedure should be clearly described in section 2 using schematic cross section, plane view etc. for the readers. Then, the device structure and size for each measurement should be described.
- “BOX” should be replaced by “gate insulator”.
- The authors investigated the device with bottom gate structure so that the heavily doped Si should be used for the bottom Si substrate as a gate.
- The interface for the channel should not be formed by the bonding.
- In Fig. 1, the authors explained the influence of positive charge in Al2O3. However, it is the well-known property of the Al2O3 gate insulator.
- In Fig. 1(b), the authors should explain the results of 600C and 800C annealing.
- In Fig. 2, the reviewer did not understand why the authors explained as “diode like” characteristics. Figure 2 shows the diode characteristic.
- Figure 3(a) should be the “drain current – gate voltage” characteristics. The authors should show the subthreshold characteristics to discuss the interface property.
- In Fig. 4, the authors should explain why the source-gate and drain-gate characteristics were compared.
- In Fig. 5(a), the authors should explain what the 1st point is. The difference of line colors should also be explained for the readers.
- In Fig. 5(a), the authors should explain why the current is electron current although the p-channel MOSFET was fabricated.
- In Fig. 6, the authors should describe the drain voltage value.
- In Fig. 6, the authors should explain why the drain current increased at the positive gate voltage region.
Author Response
Please see the attachment

This manuscript is a resubmission of an earlier submission. The following is a list of the peer review reports and author responses from that submission.
Round 1
Reviewer 1 Report
Popov et al. comprehensively studied the electrical properties of the semiconductor-insulator-semiconductor stack with hafnia-alumina nanolaminate. Various stacks have been studied, and various conduction mechanisms were considered in their study. However, the authors could not provide solid evidence for justifying their hypothesis. Therefore, the reviewer cannot recommend the publication of the manuscript in Nanomaterials. The detailed comments are attached below.
- Especially, the authors argued that the ferroelectricity could be observed in their hafnia-alumina nanolaminate, but the evidence was only the hysteresis in the capacitance-voltage curve, which cannot be sufficient evidence for proving ferroelectricity. There is no structural analysis, although the formation of specific non-centrosymmetric crystalline phase should be required. Moreover, there is no experimental evidence based on typical analysis for ferroelectrics: polarization-voltage curves, piezoelectric force microscopic images, and so on.
- The studies on the conduction mechanism also should be improved. To analyze the conduction mechanism of charge carriers, the temperature-dependent changes in current density-voltage characteristics should be experimentally examined. In this study, nonetheless, the temperature-dependent conduction behavior was not studied. The optical dielectric constant which can be taken from the temperature-dependent electrical characterization should be checked to prove the conduction mechanism.
- There are many typos and grammatical errors, which MUST be rechecked. 1) Check the subscripts and superscripts in the overall manuscript. 2) SCR was defined as the space charge region, so "SCR region" should be corrected. 3) The grammars in the overall manuscript should be carefully rechecked.
Reviewer 2 Report
In this study, the authors report on diode-like currents through ALD deposited HfO2 / Al2O3 / HfO2… nanolayers with rectification as high as 1000. The observed leakage currents are low. At low fields, thermionic emission is identified as the dominant electron transport mechanism, where Poole-Frenkel (PF) mechanism dominates at large E values. Overall the conclusions are well supported by the experimental evidence.
Some minor technical comments are as below:
- Why drain-gate current characteristics of pseudo-MOSFET after RTA at 750C are highly asymmetric in bias? and becomes symmetric after RTA of 850C (Figs. 8 a,b). Is there an explanation for this behavior?
non-technical comments:
- Authors should improve the readability of the manuscript. In its current form, it is very difficult to understand and follow.
- There are lot of acronyms used through the manuscript…please define these. (e.g. in the abstract RTA FEOL is not defined.._
Reviewer 3 Report
Review
Title: Diode-like current leakage and ferroelectric switching in silicon SIS structures with hafnia-alumina nanolaminates
Major problems
- The manuscript is incomplete : the main issues of this paper is promiscuous
- Lack of explanation for relationship between sample split and issues : the sample splits is too various to explain each issues
- Lack of evidence to support the thesis
1) Absence of control variables : In figure 4, there are is no control variables. In detail, Even if the thickness of oxide layer is different, author just suggest raw data. It is same as absence of control variables for the other data
2) Absence of explanation in charge transfer mechanism : generally, when we suggest charge transfer mechanism, we confirm the tunneling mechanism by using tunneling fitting data following each tunneling equations. There is no fitting data about tunneling mechanism.
3) Lack of verification in Ferroelectricity : There is well known method such as PUND or pulse I-t. please cross-check by using those methods.
Minor problems
- Naming of graph line is incomplete.
- Please suggest SIMS resolution : it seems that thickness of the oxide layer is over 50nm
Etc…
Overall, this manuscript was not completed, and the main issues were promiscuous. In addition, experiment data is not sufficient to prove the issues.
Reviewer 4 Report
This paper entitled “ Diode-like current leakage and ferroelectric 2 switching in silicon SIS structures with hafnia-alumina nanolaminates” described the MOS diode and MOSFET characteristics fabricated by the bonding process with high-k HfO2/Al2O3 and ferroelectric HfAlO2. However, the explanation of the fabrication process and experimental results are not clear, and some of the discussion and explanation for the obtained result are not correct. Therefore, the reviewer thinks the major revisions are necessary. The comments for the authors are listed below.
- Experimental procedure should be clearly described in section 2 using schematic cross section, plane view etc. for the readers. Then, the device structure and size for each measurement should be described.
- “BOX” should be replaced by “gate insulator”.
- The authors investigated the device with bottom gate structure so that the heavily doped Si should be used for the bottom Si substrate as a gate.
- The interface for the channel should not be formed by the bonding.
- In Fig. 1, the authors explained the influence of positive charge in Al2O3. However, it is the well-known property of the Al2O3 gate insulator.
- In Fig. 1(b), the authors should explain the results of 600C and 800C annealing.
- In Fig. 2, the reviewer did not understand why the authors explained as “diode like” characteristics. Figure 2 shows the diode characteristic.
- Figure 3(a) should be the “drain current – gate voltage” characteristics. The authors should show the subthreshold characteristics to discuss the interface property.
- In Fig. 4, the authors should explain why the source-gate and drain-gate characteristics were compared.
- In Fig. 5(a), the authors should explain what the 1st point is. The difference of line colors should also be explained for the readers.
- In Fig. 5(a), the authors should explain why the current is electron current although the p-channel MOSFET was fabricated.
- In Fig. 6, the authors should describe the drain voltage value.
- In Fig. 6, the authors should explain why the drain current increased at the positive gate voltage region.